# Retinal Development in a Precocial Bird Species, the Quail (*Coturnix coturnix*, Linnaeus 1758)

**DOI:** 10.3390/cells12070989

**Published:** 2023-03-23

**Authors:** Guadalupe Álvarez-Hernán, José Antonio de Mera-Rodríguez, Violeta Calle-Guisado, Gervasio Martín-Partido, Joaquín Rodríguez-León, Javier Francisco-Morcillo

**Affiliations:** 1Área de Biología Celular, Departamento de Anatomía, Biología Celular y Zoología, Facultad de Ciencias, Universidad de Extremadura, 06006 Badajoz, Spain; 2Área de Anatomía y Embriología Humana, Departamento de Anatomía, Biología Celular y Zoología, Facultad de Medicina, Universidad de Extremadura, 06006 Badajoz, Spain

**Keywords:** altricial, cell differentiation, immunohistochemistry, precocial, quail, retina, retinogenesis

## Abstract

The quail (*Coturnix coturnix*, Linnaeus 1758), a notable model used in developmental biology, is a precocial bird species in which the processes of retinal cell differentiation and retinal histogenesis have been poorly studied. The purpose of the present research is to examine the retinogenesis in this bird species immunohistochemically and compare the results with those from previous studies in precocial and altricial birds. We found that the first PCNA-negative nuclei are detected at Stage (St) 21 in the vitreal region of the neuroblastic layer, coinciding topographically with the first αTubAc-/Tuj1-/Isl1-immunoreactive differentiating ganglion cells. At St28, the first Prox1-immunoreactive nuclei can be distinguished in the vitreal side of the neuroblastic layer (NbL), but also the first visinin-immunoreactive photoreceptors in the scleral surface. The inner plexiform layer (IPL) emerges at St32, and the outer plexiform layer (OPL) becomes visible at St35—the stage in which the first GS-immunoreactive Müller cells are distinguishable. Newly hatched animals show a well-developed stratified retina in which the PCNA-and pHisH3-immunoreactivies are absent. Therefore, retinal cell differentiation in the quail progresses in the stereotyped order conserved among vertebrates, in which ganglion cells initially appear and are followed by amacrine cells, horizontal cells, and photoreceptors. Müller glia are one of the last cell types to be born. Plexiform layers emerge following a vitreal-to-scleral gradient. Finally, our results suggest that there are no significant differences in the timing of different events involved in retinal maturation between the quail and the chicken, but the same events are delayed in an altricial bird species.

## 1. Introduction

This study of the visual system provides an insight into neural processes in general. The retina constitutes an excellent model to examine an isolated neural network that is separated from the rest of the central nervous system (CNS), and to investigate the basic concepts of developmental biology. Abundant studies have shown that numerous genes, transcription factors, and neurotrophic factors are involved in the differentiation and formation of the vertebrate retina, acting at specific stages of the development [1,2,3,4,5,6]. The retinal neurons and Müller glia are generated in sequential but overlapping waves [7]. Birthdating analysis of retinal cell types has revealed a highly conserved temporal sequence of generation of the six neuronal and one glial cell types, although some slight variations in this sequence are seen in some vertebrate species [8].

In the case of an avian embryo, most of the studies on retinal development have been conducted on chickens (*Gallus gallus*, Linnaeus 1758) [9,10,11,12], a precocial bird species. Currently, the chicken retina is being recognized as one of the most suitable tools to study molecular mechanisms that govern neural development and connectivity [13,14,15,16,17,18,19,20,21,22,23], but also a wide range of ocular diseases and conditions [24]. In contrast, relatively little is known of how the basic events of retinal differentiation, such as neurogenesis, axon outgrowth, histogenesis, and cell death, fit into the varying developmental durations during the embryonic period in other bird species. In this sense, important differences are found in the timing of the development of various visual system morphogenesis features, and the differentiation between precocial and altricial birds. The morphogenesis of the optic vesicle and optic cup, the onset of differentiation of ganglion cells, photoreceptors, and horizontal cells, and the emergence of the plexiform layers occur at earlier stages in *G. gallus* than in altricial bird embryos [19,25,26,27]. Furthermore, the different waves of programmed cell death during visual system morphogenesis and retinogenesis is completely restricted to the embryonic period in precocial birds [28,29], while the highest incidence of programmed cell death in the retina of altricial bird species occurs in the post-hatching period [30]. At hatching, the precocial retina presents the multi-layered organization, is devoid of proliferative activity, and shows morphologically differentiated cells [1,19,20,27,31,32]. However, abundant features of immaturity are found in the retina of altricial hatchlings, with poorly developed plexiform layers, abundant proliferative activity, and the presence of morphological immature cell types [19,25,26,27,32].

The precocial quail (*Coturnix coturnix*, Linnaeus 1758) has been used as a popular animal model for developmental biology [33,34,35], but some aspects of its visual system development have been poorly explored. Some studies show the developmental program of microglia in the quail embryo retina [36,37] and the developmental roles of microglia in ontogenetic cell death that occur during the embryonic period [38]. Using the chick/quail chimeric retinospheroid technology as a model of retinogenesis, it has been shown that Müller glia actively stabilize cells within their columns during development [39]. More recently, the chronotopographical distribution of mitotic figures during quail retinogenesis and the possible relationship between the appearance of non-apical mitoses and horizontal cell differentiation have been described in detail by our group [27]. However, data on the timing of retinogenesis in this precocial bird species is limited. Here, we describe the process of cell differentiation and histogenesis in the retina of *C. coturnix* by using classical histological and immunohistochemical techniques. For this purpose, we use markers of cell proliferation and cell differentiation, and discuss the results with those described in other altricial and precocial bird species. These results could help to better understand the variability in the chronology of specific developmental features during avian ontogeny.

## 2. Materials and Methods

### 2.1. Animals and Tissue Processing

A total of 40 embryos of *C. coturnix* were used in the present study (Table 1). All animal manipulations were performed in accordance with the National and European legislation (Spanish Royal Decree RD53/2013 and EU Directive 86/609/CEE as modified by 2003/65/CE, respectively). Experimental protocols were approved by the Bioethics Committee for Animal Experimentation at the University of Extremadura (Ref. 264/2019). Eggs were incubated within a rotating egg incubator (Masalles S.A., Barcelona, Spain). The incubator was maintained at 37.5 °C and 80–90% humidity. The stages (St) described by [40] were taken as a reference to establish the degree of development. Embryos were fixed overnight at 4 °C with paraformaldehyde (PFA) 4% in a phosphate-buffered solution (PBS; 0.1 M, pH 7.4). Samples were cryoprotected with PBS-sucrose (15%) overnight at 4 °C, then soaked in embedding medium, frozen onto sectioning blocks and stored at −80 °C. Cryosections of 20µm thickness were obtained in the frontal plane and thaw-mounted on SuperFrost^®^ plus slides (Menzel-Glässer, Braunschweig, Germany), air dried and stored at −20 °C.

### 2.2. Immunohistochemistry

All sections were subjected to an antigen retrieval process with citrate buffer (pH 6.0) at 90 °C for 30 min. The slides were allowed to cool at room temperature (RT) for 20 min and then washed several times in 0.1% Triton-X-100 (Sigma-Aldrich, Saint-Louis, MO, USA) in PBS (PBS-T), each one for 10 min and in 0.2% gelatin, 0.25% Triton-X-100 in PBS (PBS-G-T) for 10 min. Sections were pre-blocked in 0.2% gelatin, 0.25% Triton-X-100, and Lys 0.1 m in PBS (PBS-G-T-L) for 1 h at RT in a humidified chamber. The dilutions and sources of primary and secondary antibodies are listed in Table 2. Sections were incubated with the primary antibody overnight at RT in a humidified chamber. The slides were rinsed several times in PBS-T and PBS-G-T, each one for 10 min, and then incubated with the secondary antibody for 2 h at RT in darkness in a humidified chamber. Sections were washed twice in PBS-T in darkness at RT for 10 min and then incubated with DAPI (4′,6-diamino-2-phenylindole, Sigma-Aldrich, Saint-Louis, MO, USA) for 10 min at RT in darkness, followed by several washes in PBS in darkness. Slides were mounted with Mowiol (polyvinyl alcohol 40–88, Fluka, Charlotte, NC, USA).

### 2.3. Image Acquisition and Processing

Immunolabeled retinal sections were observed using a Nikon Eclipse-80i microscope (Nikon, Tokio, Japan) equipped with bright field and fluorescence and photographed using an ultra-high-definition Nikon DXM 1200F (Nikon, Tokio, Japan). The preparation of graphics for publication was undertaken in Adobe Photoshop (v.CS4, Adobe Systems, San Jose, CA, USA).

## 3. Results

Retinal histogenesis in *C. coturnix* embryos was carefully examined on cryosections obtained from specimens at St19 (embryonic day 3.5, E3.5) to St46, which was the hatching day (postnatal day 0, P0). The antibodies tested in this present work have been widely used in immunohistochemical studies of retinogenesis in different groups of vertebrates, in order to analyze the different aspects of proliferative activity and cellular characterization. All of these antibodies show similar expression patterns in the quail retina to that found in the retina of other vertebrates, including birds, suggesting an interspecific conservation of the corresponding epitopes.

At St19, the *C. coturnix* retina shows an undifferentiated neuroepithelium with PCNA expression in all neuroepithelial cells, and without immunoreactivity against differentiating markers (not shown). The first PCNA-negative nuclei appear at St21 in the vitreal surface of the central region of the neuroblastic layer (NbL) (not shown), coinciding topographically with the first αTubAc-immunoreactive cells (Figure 1A–H) and the onset of expression of Isl1 in the nuclei of differentiating neurons (Figure 1I–N). αTubAc-immunopositive radial processes are also observed in the NbL (Figure 1A–H). During these early stages, abundant pHisH3-positive mitotic figures are detected in the scleral surface of the presumptive neural retina (Figure 1A–H).

At St24, PCNA-negative nuclei can be seen in the vitreal surface of NbL (Figure 2B,C). Abundant postmitotic cells are labeled with antibodies against β-tubulin class III (TUJ1 antibody) in the central region of the retina (Figure 2D–F), coinciding topographically with the PCNA-negative nuclei. The peripheral region shows a decrease in the TUJ1-immunoreactivity, suggesting a delay in the differentiation process in this region (Figure 2G–I). The TUJ1 antibody also identifies migratory neuroblasts in the NbL (Figure 2D–I).

At St28, most of the PCNA-negative nuclei (Figure 3A–C,A′–C′) coincide topographically with abundant αTubAc- (Figure 3D–G), Isl1-(Figure 3H–J), CR/TUJ1-(Figure 3K–N,K′–N′), and Prox1-immunoreactive elements (Figure 3O–Q). A few PCNA-immunonegative nuclei can also be observed in the scleral surface of the NbL (Figure 3A–C), the region where the first visinin-positive immature photoreceptors are detected (Figure 3R–T). At this stage, abundant pHisH3-positive mitotic figures are still distinguishable in the scleral surface of the NbL (Figure 3D–G).

The DAPI staining reveals the emergence of the inner plexiform layer (IPL) at St32 (Figure 4). The appearance of the IPL is confirmed with the antibody against synaptic vesicles type 2 (SV2), that identifies functional synapses externally to the presumptive GCL (Figure 4A–C). Most of the cells located in the presumptive GCL are labeled with antibodies against CR/TUJ1 (Figure 4D–G) and Isl1 (Figure 4H–J). However, scarce TUJ1-labeled cells with an elongated shape and weaker staining are found in the oNbL (Figure 4D–G). The nuclei of these cells are also immunostained with Isl1 (Figure 4H–J). Most of the Prox1-immunoreactive nuclei are located in the presumptive GCL, but some of them are distinguishable in the inner surface of the oNbL (Figure 4K–M). Abundant cells appear strongly immunolabeled in the scleral surface of the oNbL with antibodies against visinin (Figure 4N–P). These cells show an inner segment that contains the nucleus and a long outer segment (Figure 4N′–P′).

At St35, PCNA-immunoreactive cells are restricted to the external half of the INL, whereas only a few pHisH3-positive mitotic figures can be detected in the central region of the retinal tissue (Figure 5A–D). CR- and TUJ1-immunorectivity is detected in the GCL and in amacrine cell subpopulations (Figure 5E–H). Both markers colocalize in some ganglion cells (Figure 5E′–H′). Isl1-immunoreactivity is found in the nuclei of cells located in the GCL, amacrine cell layer, and horizontal cell layer (Figure 5I–L). At this stage, Prox1-immunoreactivity is almost restricted to the horizontal cell layer (Figure 5I–L) and some of these Prox1-immunostained horizontal cells also express Isl1 (Figure 5I′–L′). 

At this stage, DAPI staining reveals the emergence of the outer plexiform layer (OPL) (Figure 5 and Figure 6) that appears weakly immunostained with anti-SV2 antibody (Figure 6A–C). Moreover, visinin-immunoreactivity is intense in the ONL (Figure 6D–F), and labeled photoreceptors have a well-developed outer segment (Figure 6D′–F′). The radially-oriented somata of the first GS-immunoreactive Müller cells appear at this stage in the central retina (Figure 6G–I).

In newly hatched animals (P0), the quail retina presents a mature aspect. PCNA immunoreactive nuclei and pHisH3-positive mitotic figures are absent from the entire retina (not shown) [19,27]. CR- and TUJ1-immunoreactivities are restricted to subpopulations of cells located in the GCL and in the amacrine cell layer (Figure 7A–D,A′–D′). The nuclei of subpopulations of ganglion, amacrine, bipolar, and horizontal cells are immunoreactive for Isl1 (Figure 7E–H). Prox1 is present in the nuclei of all horizontal cells (Figure 7E–H), and some of them are also immunoreactive for Isl1 (Figure 7E′–H′). At this age, Prox1 also labels a subpopulation of amacrine cells (Figure 7G,H, arrowheads). Parvalbumin (PV) is almost confined to subpopulations of amacrine cells, with their dendrites laminarly organized in the IPL (Figure 7I–K). Weakly PV staining is also found in the OPL and in some cells located in the outer surface of the GCL (Figure 7I–K). The plexiform layers are well developed and present strong immunoreactivity against SV2 (Figure 7L–N). Furthermore, sparse cell somata, located in the amacrine cell layer, are also immunolabelled with this antiserum (Figure 7L–N). Visinin-immunoreactivity can be seen in morphologically mature photoreceptors (Figure 7O–Q). Finally, GS-immunolabelling identifies Müller cells which stretch across the entire breadth of the retinal tissue (Figure 7R–T). The immunoreactivity is particularly intense in the medial region of the INL, where the Müller cell somata are located (Figure 7R–T).

## 4. Discussion

In this work, we show the cell differentiation dynamics and the timing of the emergence of plexiform layers in the developing retina of *C. coturnix*, a precocial bird species. These data are compared with results previously reported in other avian embryos, including precocial and altricial bird species.

### 4.1. Retinal Histogenesis and Cell Differentiation in C. coturnix

#### 4.1.1. Neurochemical Profiles in the Undifferentiated and Laminated Retina

Knowledge of protein expression during cell differentiation is pivotal to gaining a better understanding of the events that drive retinal neurogenesis in vertebrates [41]. PCNA antigen is known to be expressed in essentially all retinal progenitor cells in the developing and mature avian retina [19,25,42,43]. In the early developing quail retina, PCNA is detected in all neuroepithelial cells early stages and, at the onset of retinogenesis, excluded from cells located in the inner surface, coinciding topographically with the first Isl1-/αTubAc-immunoreactive cells. Isl1 distribution in the developing retina of vertebrates has been extensively studied (for a review, see [1]), and is known to be expressed in postmitotic ganglion and amacrine cells in the early stages of retinogenesis [44,45,46]. In the case of the distribution of αTubAc during early stages of retinogenesis, it labels the pioneer ganglion cell axons in zebrafish [47], chickens [20] and mammals [48]. Therefore, ganglion cells are the first types of neurons to differentiate in the developing quail retina, coinciding with results described in the developing chicken [1,20,49,50,51,52,53] and zebra finch retina [25,26]. 

At more advanced stages (St28), a higher amount of Isl1-/αTubAc-/CR-immunoreactive ganglion cells is detected in the vitreal surface of the NbL, coinciding topographically with abundant PCNA-negative nuclei. Furthermore, at this stage, the first Prox1 positive nuclei are also detected near the vitreal surface. Prox1 is considered to be a pan-horizontal cell marker, but is also expressed in a subpopulation of amacrine cells (Dyer et al., 2003). The initial expression for this transcription factor coincides with the onset of amacrine and horizontal cell differentiation in the chicken [12,54,55], but also in the quail [27]. At this embryo stage, the first PCNA-negative nuclei are also detected in the scleral surface of the quail NbL, coinciding again with the onset of expression of visinin, a calcium binding protein expressed in early differentiating photoreceptors in the avian retina [20,26,56,57]. Therefore, ganglion cells, amacrine cells, horizontal cells and photoreceptors are the first classes of neurons produced in the quail retina, coinciding with the order of cell differentiation described in the chicken [9,49] and the zebra finch [25,26].

At St35, PCNA-immunoreactivity is restricted to the external half of the INL in the developing quail. This late proliferation seems to give rise mainly to bipolar cells, some photoreceptor cells, and Müller glia, the latest cell types that become differentiated in the avian retina [9,25,26,49]. 

In sum, the neurochemical profiles and the order of cell differentiation in the developing quail retina are similar to those described in other bird species.

#### 4.1.2. Emergence of the Plexiform Layers

The emergence of the plexiform layers has been monitored immunohistochemically with the anti-SV2 antibody, which specifically identifies the appearance of functional synapses in the avian retina [20,25,58]. Immunohistochemical staining reveals that the formation of the OPL is delayed with respect to the emergence of the IPL in the developing quail retina, suggesting a vitreal-to-scleral gradient of differentiation in this bird species. These patterns of cell differentiation seem to be conserved in the avian retina [25,26,31,59,60,61,62,63]. 

The vitreal-to-scleral gradient of histogenesis is not a universal feature of the vertebrate retina. A similar inner-to-outer pattern of cell differentiation and histogenesis has been described in the developing retina of cartilaginous fish [8,64,65,66] and mammals (Reese et al., 1996; Rapaport et al., 2004). However, the emergence of both plexiform layers occurs simultaneously in the developing retina of bony fish [67,68,69] and reptiles [44]. 

#### 4.1.3. Timing of Retinal Development

In both precocial and altricial species, embryos need different periods of time to overpass different developmental stages [70]. Some authors show that altricial bird species are characterized by a rapid early development of “supply” organs, such as digestive organs, but precocial bird species exhibit a more rapid early development of other “demand” organs, such as brain and sense organs [71,72]. Our data show that, in the case of the visual system, the onset of early retinal cell differentiation in *C. coturnix* is detected immunohistochemically at 96 h of incubation, while in the *T. guttata* (altricial bird species) retina cell differentiation starts at 108 h of incubation [25,26] (see Table 3). In general, the onset of expression of cell markers used in the present study occurs earlier in the quail retina than in the altricial retina (see Table 3). Furthermore, other ontogenetic changes such as the emergence of the plexiform layers occurs much later in the *T. guttata* retina [25,26] than in the quail retina (IPL: E8.5 vs E6.5; OPL: E9 vs E8). Additionally, other studies have demonstrated that the disappearance of apical and non-apical mitoses, and the incidence of cell death, occur in the embryonic period in precocial bird species such as the quail [27,38] and in the chick [28], while in the *T. guttata*, mitotic activity and cell death are detected in the retina of post-hatched animals [30]. Therefore, there is a delay in retinal tissue maturation in altricial embryos in comparison with the quail.

Concerning the comparison between the quail and the chicken, our results also show slight variations in the timing of retinogenesis between these precocial bird species. Early cell differentiation of ganglion cells, amacrine cells, and photoreceptors occur earlier in the chicken retina than in the quail retina [1,19,49,50,74,79,80] (see Table 3), indicating faster ontogeny during the early stages. However, the first differentiated horizontal cells and the plexiform layers appear earlier in the quail than in the chicken [27,75] (see Table 3). 

Therefore, there are only minimal differences between developmental features in quail and chick embryos during the early stages of development, in agreement with results previously reported [35,76].

#### 4.1.4. Retinal Structure at Hatching

Differences among the altricial/precocial spectrum, regarding the functional maturity of the tissues are more evident in newborn/hatchling vertebrates [74,75,76,77,78,79,80]. Hatching *C. coturnix* have open lids and fully developed retinas with all of their layers well established, in agreement with previous studies realized in precocial birds [19,20,32,79]. PCNA- and pHisH3-immunoreactivity is absent from the entire retina in this bird species (present study; [27]), and visinin-immunoreactive photoreceptors are morphologically mature with a well-developed outer segment. 

On the contrary, altricial newly hatched animals are blind and with their eyes closed [19,32]. The retina of altricial hatchlings shows the multilayered structure, although features of immaturity are observable, such as GCL containing many layers of nuclei and very thin plexiform layers [25,32]. Abundant PCNA-immunoreactivity is detected in the outer half of the INL and in the peripheral-most retina [19,25]. Furthermore, many pHisH3-immunoreactive apical and non-apical mitoses can also be found in the retina of newly hatched animals [27]. Moreover, photoreceptors show an immature morphology with poorly developed outer segments [25,32]. Therefore, the retina of *C. coturnix* is well developed at hatching since most of the retinal maturation occurs in the embryonic period.

## 5. Conclusions

This paper has shown that the quail retina is fully developed at hatching, as has been described in other precocial bird species. It is clearly shown that the order of different events involved in quail retinal maturation is the same during development compared to other bird species, but their specific timing differs between precocial and altricial bird species. Together, our results expand our current understanding of the timing and cellular differences that regulate patterns of avian retinal growth and maturation and provides a better understanding of the evolution of avian altriciality and precociality.

## Figures and Tables

**Figure 1 cells-12-00989-f001:**
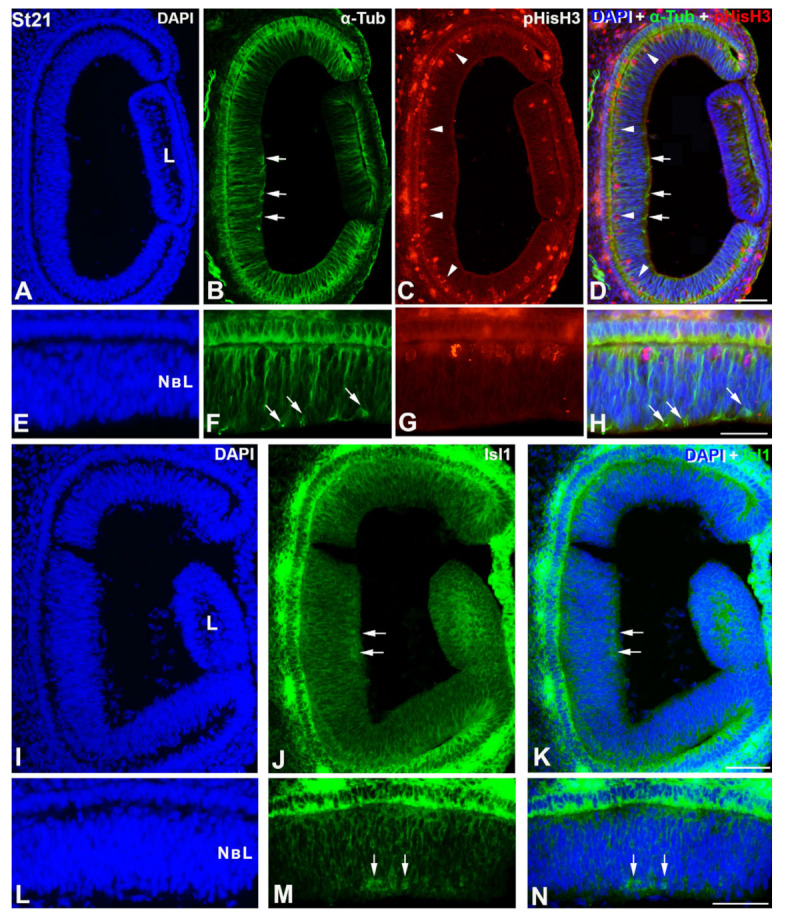
Onset of cell differentiation in the retina of *C. corturnix* at St 21- of development. DAPI staining reveals that quail retina is composed of a NbL at this stage (**A**,**E**,**I**,**L**). The first α-Tub-positive cells are detected at this stage in the inner surface of the neural tissue (arrows in **B**,**D**,**F**,**H**). Radially oriented α-Tub-immunoreactive elements are also observed (**B**,**D**,**F**,**H**). pHisH3 antibody shows abundant mitotic cells distributed throughout the scleral surface of the NbL (arrowheads in (**C**,**D**,**G**,**H**)). Isl1-immunoreactivity is observed in cell nuclei located in the inner surface of the NbL. These Isl1-positive nuclei are restricted to the central region of the undifferentiated retina (arrows in (**J**,**K**,**M**,**N**)). L, lens; NbL, neuroblastic layer. Scale bars: 100 μm.

**Figure 2 cells-12-00989-f002:**
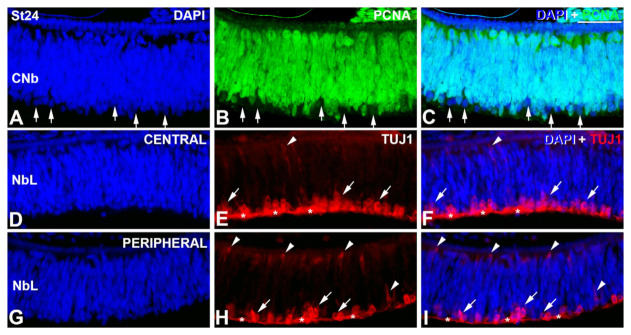
PCNA (**A**–**C**) and TUJ1 (**D**–**I**) immunoreactivity in retinal cryosections of *C. coturnix* at St 24–25 of development. DAPI staining reveals that quail retina is composed of a NbL at this stage (**A**,**D**,**G**). PCNA-negative nuclei point to those cells that have started the differentiation process (arrows in (**A**–**C**)). Central (**D**–**F**) and peripheral (**G**–**I**) regions of the undifferentiated retina show strong TUJ1-immunoreactivity in both migratory neuroblasts (arrowheads in (**E**,**F**,**H**,**I**)), and ganglion cell axons in the optic fiber layer (asterisks). Notice that the number of TUJ1 immunoreactive elements located in the inner surface seems to be higher in the central region. NbL, neuroblastic layer. Scale bars: 100 μm.

**Figure 3 cells-12-00989-f003:**
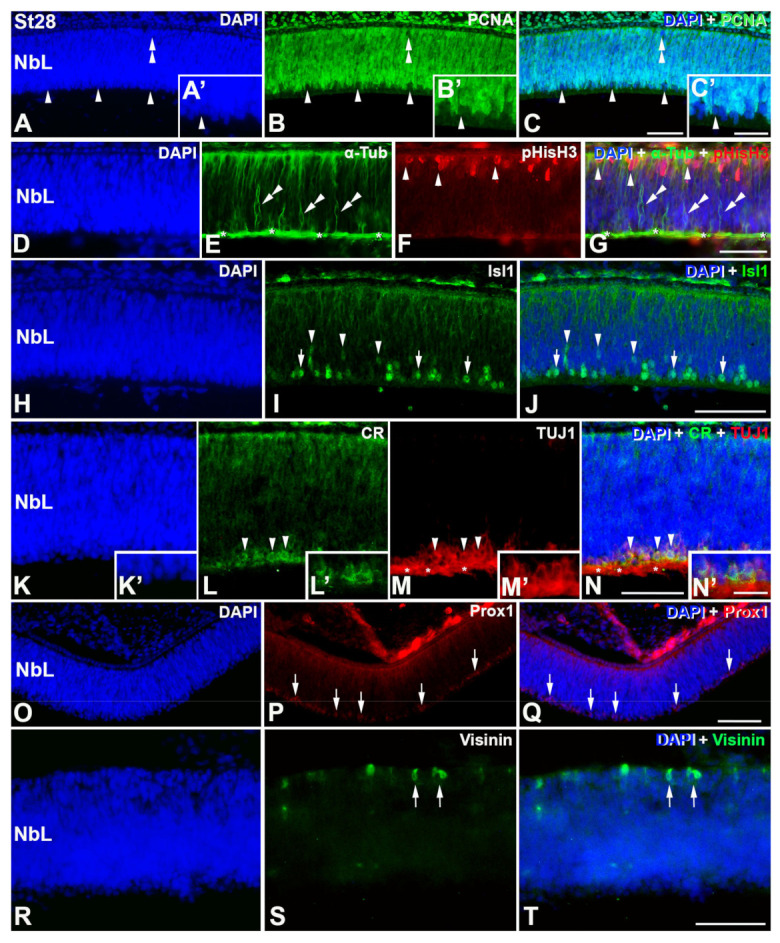
Detection of markers of cell differentiation in retinal cryosections of *C. coturnix* at St28 of development. DAPI staining reveals that quail retina is composed of a NbL at this stage (**A**,**D**,**H**,**K**,**O**,**R**). PCNA-immunonegative nuclei are found in the inner surface of the NbL (arrowheads in (**A**–**C**, **A′**–**C′**), but also in the outer surface of the undifferentiated retina (double arrowheads in (**A**–**C**)). Strong α-Tub-immunoreactivity is detected in the presumptive OFL (asterisks in (**E**,**G**)) and in the cytoplasm of migratory neuroblasts (double arrowheads in (**E**,**G**)). pHisH3 immunoreactive mitotic figures are abundant in the scleral surface of the NbL (arrowheads in (**F**,**G**)). Islet1 immunosignal is located in both the nuclei of migratory neuroblasts (arrowheads in (**I**,**J**)) and cell nuclei located in the inner surface of the NbL (arrows in (**I**,**J**)). Double immunostaining reveals that some TUJ1 immunoreactive cells located in the inner surface of the NbL also express CR (arrowheads in (**L**–**N**) and **(K’**–**N’**)). The presumptive OFL is also strongly TUJ1-immunoreactive (asterisks in **M**,**N**). Prox1 is first detected at this stage in cell nuclei located close to the vitreal surface of the NbL (arrows in (**P**,**Q**)). Sparsely visinin-immunoreactive cells are firstly distinguishable in the outer surface of the NbL (arrows in (**S**,**T**)). NbL, neuroblastic layer. Scale bars: 100 μm in A-T; 30 μm in (**A′**,**C′**,**K′**–**N′**).

**Figure 4 cells-12-00989-f004:**
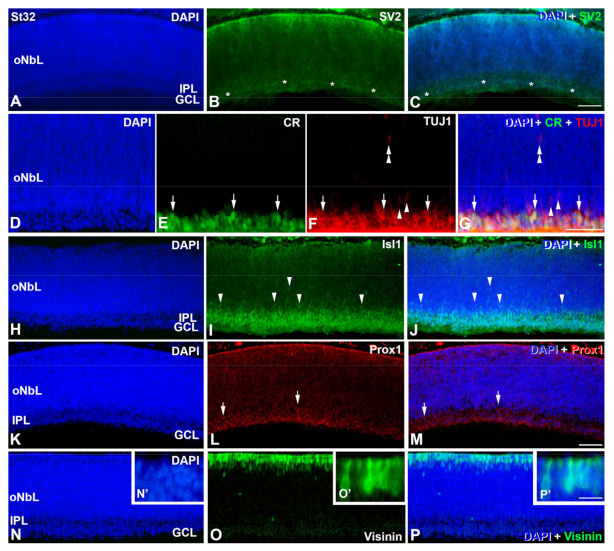
Immunodetection of different cell markers in retinal cryosections of *C. coturnix* at St32 of development. DAPI staining reveals an incipient emergence of the IPL (**A**,**E**,**H**,**K**,**N**). A faint immunostaining against SV2 is detected in the developing IPL (asterisks in (**B**,**C**)). Most of the cells located in the GCL are doubly labeled for CR and TUJ1 (arrows in (**D**–**G**)). Single TUJ1-labeled cells are also detected in the pGCL (arrowheads in (**F**,**G**)) and in more external regions (double arrowheads in (**F**,**G**)). A few rows of Isl1-positive nuclei can be observed in the pGCL (**I**,**J**). Sparse Isl1-immunopositive nuclei are also observed in more external regions (arrowheads in (**I**,**J**)). Prox1-immunostaining localizes in the innermost region of the oNbL and in the GCL (arrows in (**L**,**M**)), but also in more external regions of the oNbL (double arrowheads). Visinin-immunoreactive cells are mainly located in the more external region of the oNbL, but sparse labeled cells can be distinguished in more internal regions (arrowheads in (**O**,**P**)). Visinin-positive cells clearly show a cell somata with a short external process (**O′**,**P′**). GCL, ganglion cell layer; IPL, inner plexiform layer; NbL, neuroblastic layer; oNbL, outer neuroblastic layer; pGCL, presumptive ganglion cell layer. Scale bars: 100 μm in (**A**–**P**); 30 μm in (**N′**–**P′**).

**Figure 5 cells-12-00989-f005:**
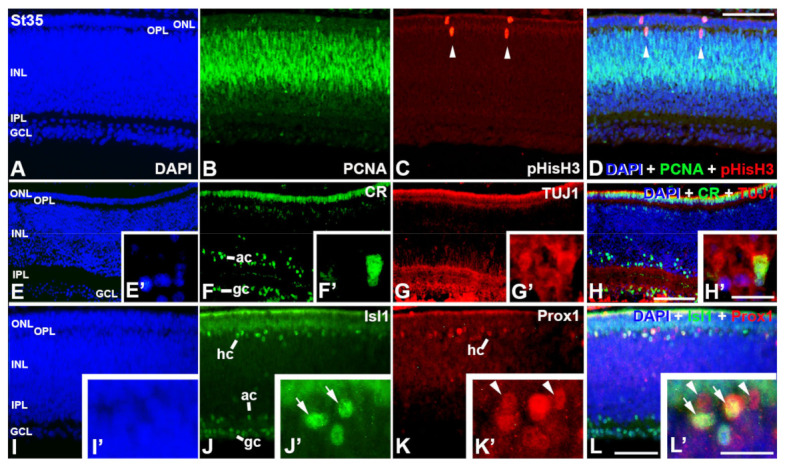
Cell markers in retinal cryosections of *C. coturnix* at St35 of development. DAPI staining reveals the multilayered structure of the quail retina (**A**,**E**,**I**). In the central retina, PCNA-positive proliferating cells are mainly restricted to the outer half of the INL (**B**,**D**). pHisH3-immunoreactive mitotic figures are sparse and are mainly found in the outer INL and in the ONL (arrowheads in (**C**,**D**)). Abundant ganglion and amacrine cells are immunoreactive against CR (**F**,**H**) and TUJ1 (**G**,**H**). A few ganglion cells co-express these markers (**F′**–**H′**). Islet1-immunoreactivity identify ganglion, amacrine and horizontal cells (**J**,**L**). Prox1-positive nuclei are found in the presumptive horizontal cell layer (**K**,**L**) and some of them also express Isl1 (**J′**–**L′**). GCL, ganglion cell layer; INL, inner nuclear layer; IPL, inner plexiform layer; ONL, outer nuclear layer; OPL, outer plexiform layer. Scale bars: 150 μm in A-L; 30 μm in (**E′**–**L′**).

**Figure 6 cells-12-00989-f006:**
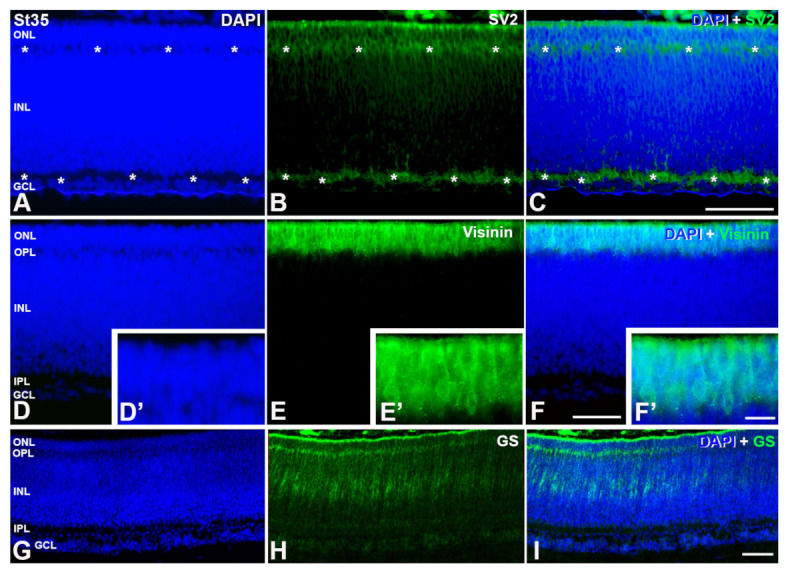
SV2 (**A**–**C**), visinin (**D**–**F**, **D′**–**F′**) and GS (**G**–**I**) immunostaining in retinal cryosections of *C. coturnix* at St 35 of development. OPL can be observed at this stage as well as IPL (asterisks in (**A**)) using SV2-antibody (asterisks in (**B**,**C**)). Visinin-immunosignals is stronger, showing a higher number of photoreceptors (**E**,**F**,**E′**,**F′**). Müller cells GS-positive are detected for first time (**H**,**I**). GCL, ganglion cell layer; INL, inner nuclear layer; IPL, inner plexiform layer; ONL, outer nuclear layer; OPL, outer plexiform layer. Scale bars: 100 μm in (**A**–**I**); 30 μm in (**D′**–**F′**).

**Figure 7 cells-12-00989-f007:**
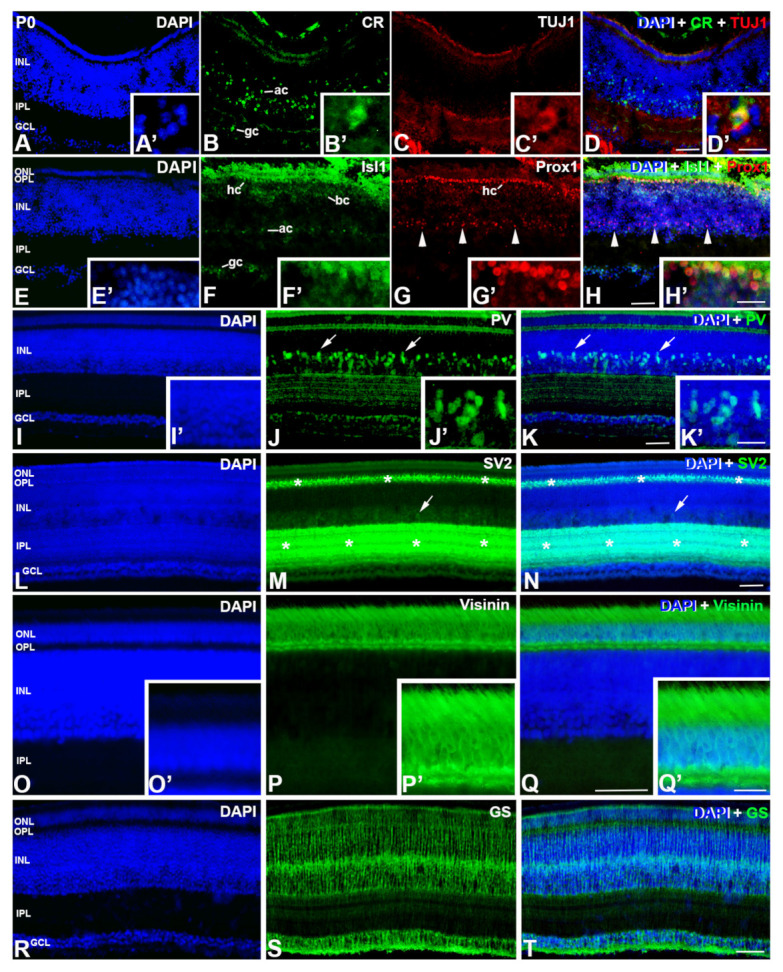
Immunoexpression of different cell markers in retinal cryosections of *C. coturnix* at P0. An increase of amacrine cells CR and TUJ1-immunoreactive is detected (**B**–**D**). Co-expression of these markers is found in some of the ganglion cells (**B′**–**D′**). Ganglion, amacrine, bipolar and horizontal cells are Isl1-positive (**F**,**H**). Prox1-immunoreactivity can be localized in horizontal cells (**G**,**H**) as well as in some horizontal cell precursors (arrowheads in (**G**,**H**)). Co-staining with Islet1 and Prox1 can be detected in a subpopulation of horizontal cells (**F′**–**H′**). A mature multilayered tissue with amacrine cells and PV-immunoreactive can be detected in the INL. (**J**,**K**,**J′**,**K′**). Plexiform layers (asterisks in (**M**,**N**)) can be localized because of the SV2 antibody as well as some amacrine cells (arrows in (**M**,**N**)). Visinin expression is detected in mature photoreceptors (**P**,**Q**) that exhibits outer segments (**P′**,**Q′**). An intense GS-immunosignal is observed in the somata and feet of Müller cells (**S**,**T**). GCL, ganglion cell layer; INL, inner nuclear layer; IPL, inner plexiform layer; ONL, outer nuclear layer; OPL, outer plexiform layer. Scale bars: 100 μm in (**A**–**T**); 50 μm in (**I′**–**K′**), (**O′**–**Q′**); 30 μm in (**A′**–**H′**).

**Table 1 cells-12-00989-t001:** Stages of development and number of *C. coturnix* embryos used in the present study.

Stage	Incubation Time	N
St19	72 h (3 days)	5
St 21	84 h (3.5 days)	6
St 24	108 h (4.5 days)	6
St 28	132 h (5.5 days)	5
St 31	6.5 days	4
St 32	7 days	4
St 35	8–9 days	4
St 42	13 days	3
St 46 (P0)	16.5 days (hatching)	3
**TOTAL**		40

**Table 2 cells-12-00989-t002:** Dilutions and sources of the primary and secondary antibodies used in the present study.

Primary Antibodies	Working Dilution	Antibody Suppliers	Cell Type Specificity
Mouse anti-β-tubulin class III monoclonal antibody (TUJ1)	1:200	Abcam (Ref. ab14545)	Neuroblasts, ganglion and amacrine cells
Mouse anti-PCNA monoclonal antibody (clone PC10)	1:200	Santa Cruz Biotechnology (Ref. sc-56)	Proliferating cells
Mouse anti-visinin monoclonal antibody (clone 7G4)	1:200	Sigma Aldrich (Ref. P0089)	Photoreceptors
Mouse anti-islet1 monoclonal antibody (clone 40.2D6)	1:50	Developmental Studies Hybridoma Bank (Ref. 40.2D6)	Neuroblasts, ganglion, amacrine, bipolar and horizontal cells
Mouse anti-glutamine synthetase monoclonal antibody	1:200	Millipore (Ref. LV1412159)	Müller cells
Mouse anti-synaptic vesicle glycoprotein 2A (SV2) monoclonal antibody	1:200	Developmental Studies Hybridoma Bank (ref. AB_2315387)	Axons and some amacrine cells
Mouse anti-acetylated α-Tubulin monoclonal antibody	1:200	Santa Cruz Biotechnology (Ref. sc-3950)	Neuroblasts, ganglion and amacrine cells
Mouse anti-Parvalbumin monoclonal antibody	1:200	Millipore (Ref. MAB1572)	Subpopulation of amacrine cells
Rabbit anti-Prox1 polyclonal antibody	1:200	Millipore (Ref. AB5475)	Amacrine, horizontal cells and horizontal cell precursors
Rabbit anti-calretinin polyclonal antibody	1:500	Swant (Ref. 7697)	Ganglion and amacrine cells
Rabbit anti-phospho Histone H3 (Ser10) polyclonal antibody	1:200	Millipore (Ref. 06-570)	Mitotic cells
Secondary antibodies	Working dilution	Antibody suppliers	
Alexa Fluor 488 goat anti-mouse IgG antibody	1:200	Molecular Probes (Ref. A11029)	
Alexa Fluor 594 goat anti-mouse IgG antibody	1:200	Molecular Probes (Ref. A11032)	
Alexa Fluor 488 goat anti-rabbit IgG antibody	1:200	Molecular Probes (Ref. A11008)	
Alexa Fluor 594 goat anti-rabbit IgG antibody	1:200	Molecular Probes (Ref. A11037)	

**Table 3 cells-12-00989-t003:** Chronology of different features of cell differentiation and histogenesis in the developing and hatched retina of quail, chicken, and zebra finch. HH, stages of Hamburger and Hamilton (1951). Asterisks correlate the results with the publications where they appeared first.

Event	*Coturnix coturnix*	*Gallus gallus*	*Taeniopygia* *guttata*	References
**Proliferative activity at hatching in the laminated retina**	Absent	Absent *	Abundant **	* [19,73] Fischer and Reh, (2000) ** [19,26]
**Ganglion cell differentiation**	PCNA-negative nuclei: St21-22 (96 h)	[3H]timidine HH13 (48 h) *	PCNA-negative nuclei: St24 (108 h) **	* [49]** [26]
	α-Tubulin expression: St21 (96 h)	Tuj1 expression: HH16 (51–56 h) *	Tuj1 expression: St24 (108 h) **	* [74]** [26]
	Isl1 expression: St22 (96 h)	Isl1 expression: HH19-20 (72 h) *	Isl1 expression: St25 (120 h) **	* [75]*S* * [1]** [26]
**Amacrine cell differentiation**	Tuj1 expression: St35 (E8)	Tuj1 expression: HH34 (E8) *	Tuj1 expression: St41 (E10) **	* [74] ** [26]
	Isl1 expression: St35 (E8)	Isl1 expression: HH34 (E8) *	Isl1 expression: St41 (E10) **	* [76]** [26]
**Photoreceptor differentiation**	PCNA-negative nuclei: St28 (132 h)	[3H]timidin: HH24 (96 h) *	PCNA-negative nuclei: St28 (132 h) **	* [49]** [26]
	Visinin expression: St28 (132 h)	Visinin expression: HH27 (120 h) *	Visinin expression: St28 (132 h) **	* [77]** [26]
**Horizontal cell differentiation**	Prox1 expression: St28 (132 h) **	Prox1 expression: HH30 (156 h)-HH32 (174 h) *	Prox1 expression: St34 (174 h) **	* [78] ** [26,27]
**IPL emergence**	SV2 expression: St31 (156 h)	SV2 expression: HH31 (E7) * AChE-immunoreactive band (E7) ** PNA-histochemistry ***	SV2 expression: St38 (E8.5) ****	* [31] ** [63] ***** [59] ** [25]
**OPL emergence**	SV2 expression: St35 (E8)	SV2 expression: HH34 (E8) *	SV2 expression: St39 (E9) **	* [31]** [25]

## Data Availability

Some or all data used during this study are available from the corresponding author by request.

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
