# Peer review of "Retinal Development in a Precocial Bird Species, the Quail (Coturnix coturnix, Linnaeus 1758)"

_cells, 2023, doi:10.3390/cells12070989_

Round 1
Reviewer 1 Report
This is a report on retinal differentiation patterns of the Japanese quail (Coturnix coturnix). Applying some relevant immunohistochemical markers, including PCNS, pHisH3, Islet-1, α-Tubulin, TUJ-1, calretinin, Prox12, visinin, allowed to follow differentiation of most major retinal cell types at 3-4 different developmental stages, the oldest one being P0 (hatching). The number of analyzed stages is small. Similarly, the spatial resolution of results is limited, since only figure 2 presents center-peripheral differences. Therefore, these results give only a brief overall picture, not providing a complete and coherent picture of quail retinal development. At times, the quality of figures appears mediocre (at least on my Mac machine), frames are blurred and out of focus (see below). In Discussion (section 4) the results are compared with retinae from chick and the altricial zebrafinch, Taeniopygia guttata. The MS is well organized and written. The list of citations includes many self citations, while other relevant citations are incomplete (note, my mentions below also remain incomplete).
Comments that require response from authors:
- l. 28: since only one altricial bird species was considered in Discussion, here it should say „in an altricial bird species“.
- l. 66: the chick-quail model has been used in a highly innovative manner to produce chimeric retinal cell reaggregates (organoids), which demonstrated functions of early Müller cells in forming and stabilizing retinal cell columns (Eur J Neurosci 7, 2277-2284, 1995). This reference is relevant here.
- Table 2: indicate cell type specificity of antibodies;
- l. 87: why is Ainsworth et al. not cited by a number? plse. be consistent & correct;
- Fig. 1: some frames are blurred, pHisH3 (C, G) not in focus, Isl-1 (J, M) staining barely visible;
- l. 145: PCNA is shown only in Fig. 2B and C, not in 2A: correct in text;
- Fig. 2B,C are blurred;
- Fig. 2E,H and F,I: what are 2-3 small white dots?, plse. explain in legend.
- l. 173: where are asterisks in frames E,G?
- Fig. 3 P,Q: Prox1-staining at arrows not visible;
- l. 177/178: double staining of TUJ1 and CR not well visible/documented;
- Fig. 4E: label „CR“ is missing in frame E;
- l. 188 and Fig. 4I,J: Isl1-staining is not restricted to presumptive GCL, as claimed in text: plse. comment;
- Fig. 4L.M: Prox1-staining is weak;
- Legend Figs. 4, 5: list of abbreviations is incomplete, e.g., pGCL, oNBL: correct;
- l. 231ff: results of Fig. 5 are not described in running text: correct;
- l. 234 „…have a well-developed outer segment…“: this statement is only barely supported by the figure frames; instead say: …is indicated that outer segments are well-developed.
- l. 247: indicate that PCNA and pHisH3 results are not shown;
- Fig. 7: most frames are blurred, e.g., C,D,M,P;
- What sections are magnified in F’,G’,H’?
- l. 319: emergence of IPL: the very first indication of IPL formation in the chick is the occurrence of cholinergic markers. Cite relevant papers, e.g., Cell Tissue Res. 286, 13-22, 1996; Dev Brain Res 100, 62-72, 1997; Cell Tissue Res. 288, 407-416, 1997; J Comp Neurol 520: 3181-93. doi: 10.1002/cne.23083.
- l. 328:: as far as I am aware, there is no more detailed 3D study of spatio-temporal differentiation of the chick retina, as was described in Dev Brain Res. 8, 223-229, 1983 and in Neuroscience 9, 931-941, 1983. They are still gold standard and should be cited, if one talks about avian spatio-temporal differentiation.
- l. 332ff: relative timing of development: I agree with the general discussion in this paragraph. However, I wonder what the total developmental time of T. Guttata is (this must be given in legend to Table 3)? Does it make sense to compare hours of development, if the total embryonic periods before hatching are very different (chick 21 days vs. 16.5 in quail). A better measure would be to give the percentage of embryonic periods.
Typos & minor comments:
l. 13: type …the immunohistochemical retinogenesis in…;
l. 20: type …layer becomes visible…;
l. 46: type …the chicken retina is being…;
l. 50: type …durations during the embryonic…;
l. 92: type … of 20μ thickness…;
l. 100: type …in 0.2% gelatine, 0.25% Triton… (be consistent in using dots vs. commas)
l. 179: type …also strongly TUJ1…
l. 181: type …sparsely visinin-…
l. 271: type …can be localized…;
l. 273: type …in somata and feet…;
l. 293: type …cells at early stages…;
Author Response
Referee 1
This is a report on retinal differentiation patterns of the Japanese quail (Coturnix coturnix). Applying some relevant immunohistochemical markers, including PCNS, pHisH3, Islet-1, α-Tubulin, TUJ-1, calretinin, Prox12, visinin, allowed to follow differentiation of most major retinal cell types at 3-4 different developmental stages, the oldest one being P0 (hatching). The number of analyzed stages is small. Similarly, the spatial resolution of results is limited, since only figure 2 presents center-peripheral differences. Therefore, these results give only a brief overall picture, not providing a complete and coherent picture of quail retinal development. At times, the quality of figures appears mediocre (at least on my Mac machine), frames are blurred and out of focus (see below). In Discussion (section 4) the results are compared with retinae from chick and the altricial zebrafinch, Taeniopygia guttata. The MS is well organized and written. The list of citations includes many self citations, while other relevant citations are incomplete (note, my mentions below also remain incomplete).
Response
Thank you very much for your comments. Yes, we know that the number of stages is small, but the description of the dynamic expression of the antigens in the developing quail retina as well as the main histogenetic events are complete. The first draft of the manuscript had 11 figures. In this first draft we also analysed St37-38 and St42, but the patterns of distribution of antigens were very similar to that observed in the P0 retina (once the OPL has emerged). The results were very repetitive from St35 onwards. Therefore, in our opinion the staining patterns in the laminated retina are summarized in the P0 figure.
Referee 1
Comments that require response from authors:
- 28: since only one altricial bird species was considered in Discussion, here it should say “in an altricial bird species“.
Response
We agree with the referee 1
- 66: the chick-quail model has been used in a highly innovative manner to produce chimeric retinal cell reaggregates (organoids), which demonstrated functions of early Müller cells in forming and stabilizing retinal cell columns (Eur J Neurosci 7, 2277-2284, 1995). This reference is relevant here.
Response
This comments are included in the revised version of the paper
Referee 1
Table 2: indicate cell type specificity of antibodies
Response
Included in the new version
Referee 1
- 87: why is Ainsworth et al. not cited by a number? plse. be consistent & correct;
Response
Corrected in the new version
Referee 1
Fig. 1: some frames are blurred, pHisH3 (C, G) not in focus, Isl-1 (J, M) staining barely visible;
Response
We agree with referee 1. We adjust some parameters in the panels. The staining of Isl1 is bare by this early stages and it is difficult to obtain better staining.
Referee 1
- 145: PCNA is shown only in Fig. 2B and C, not in 2A: correct in text;
Response
Corrected.
Referee 1
Fig. 2B,C are blurred;
Response
Improved in the revised version
Referee 1
Fig. 2E,H and F,I: what are 2-3 small white dots?, plse. explain in legend.
Response
Ganglion cell axons. Included in the revised text.
Referee 1
- 173: where are asterisks in frames E,G?
Response
The asterisks were black in the first submission. Now are white asterisks in the new version.
Referee 1
Fig. 3 P,Q: Prox1-staining at arrows not visible;
Response
The initial expression of Prox1 is weak. We have increase the contrast in the new images.
Referee 1
- 177/178: double staining of TUJ1 and CR not well visible/documented;
Response
We think that it is clearly shown in magnifications L´, M´, N´ and it is described in the legend, but we have added that “…some TUJ1 immunoreactive cells located in the inner surface of the NbL also express CR…”
Referee 1
Fig. 4E: label „CR“ is missing in frame E;
Response
It has been added.
Referee 1
- 188 and Fig. 4I,J: Isl1-staining is not restricted to presumptive GCL, as claimed in text: plse. comment;
Response
We agree with referee 1, Isl1 is not restricted to the GCL, but in the text (line 194) and in the legend (line 208) is has been described that cells located in more external regions of the differentiating retina are also immunoreactive against Isl1.
Referee 1
Fig. 4L.M: Prox1-staining is weak;
Response
We increased the contrast in the new image.
Referee 1
Legend Figs. 4, 5: list of abbreviations is incomplete, e.g., pGCL, oNBL: correct
Response
Included in the new text in figure 4. Figure 5 does not include these abbreviations in the panels.
Referee 1
- 231ff: results of Fig. 5 are not described in running text: correct;
Response
We are not sure to understand Referee 1. Fig 5 is well described in lines 216-223 (220-227 in the revised text). The figure legend is a detailed description of the figure.
Referee 1
- 234 „…have a well-developed outer segment…“: this statement is only barely supported by the figure frames; instead say: …is indicated that outer segments are well-developed.
Response
In the revised version the outer segments are clearly visible.
Referee 1
- 247: indicate that PCNA and pHisH3 results are not shown.
Response
Thank you very much for your advice. It was our mistake. Now is added to the revised version.
Referee 1
Fig. 7: most frames are blurred, e.g., C,D,M,P;
Response
We have tried to improve the images in the revised version.
Referee1
What sections are magnified in F’,G’,H’?
Response
Thank you very much for noticing, it was a mistake in the text. The cells described in the text related to the magnifications are horizontal cells. We have corrected it.
Referee 1
- 319: emergence of IPL: the very first indication of IPL formation in the chick is the occurrence of cholinergic markers. Cite relevant papers, e.g., Cell Tissue Res. 286, 13-22, 1996; Dev Brain Res 100, 62-72, 1997; Cell Tissue Res. 288, 407-416, 1997; J Comp Neurol 520: 3181-93. doi: 10.1002/cne.23083.
Response
We thank to referee 1. The references of Thangaraj et al. (2012) coincides in the onset of AChE immunostaining in the IPL (E7). This reference has been added to the Table 3 and to the text. We also added the references of Reiss et al. (1996ab) and Layer et al. (1997) to the text.
Referee 1
- 328:: as far as I am aware, there is no more detailed 3D study of spatio-temporal differentiation of the chick retina, as was described in Dev Brain Res. 8, 223-229, 1983 and in Neuroscience 9, 931-941, 1983. They are still gold standard and should be cited, if one talks about avian spatio-temporal differentiation.
Response
We thank again to referee 1. The paper of Liu et al. (1983) has been added to the table 3 and to the text.
Referee 1
- 332ff: relative timing of development: I agree with the general discussion in this paragraph. However, I wonder what the total developmental time of T. Guttata is (this must be given in legend to Table 3)? Does it make sense to compare hours of development, if the total embryonic periods before hatching are very different (chick 21 days vs. 16.5 in quail). A better measure would be to give the percentage of embryonic periods.
Response
Thank you very much for the advice of referee 1. However we think that the percentage is not a good measure since a P0 zebrafinch is less developed than chick/quail P0 specimens. In fact, a P0 zebrafinch is less developed than a E16 chicken/quail. We compare the velocity of developmental events from the lay of the egg, as has been described in other studies that compare altricial/precocial heterochrony during development (see Murray et al., 2013, J. Morphol. 274 (1090-1110)) or compare some aspects of embryo development using a morphological feature to unify embryo stages in different bird species (number of somites, length of the limb) (Blom and Lilja, 2005, Zoology 108: 81-95). Here we compare the timing in the onset of the emergence of the IPL, OPL, ganglion cell differentiation…in different bird species (altricial and precocial) and see that precocial bird species reach every developmental event in less time than the altricial bird.
Referee 1
Typos & minor comments:
- 13: type …the immunohistochemical retinogenesis in…;
- 20: type …layer becomes visible…;
- 46: type …the chicken retina is being…;
- 50: type …durations during the embryonic…;
- 92: type … of 20μ thickness…;
- 100: type …in 0.2% gelatine, 0.25% Triton… (be consistent in using dots vs. commas)
- 179: type …also strongly TUJ1…
- 181: type …sparsely visinin-…
- 271: type …can be localized…;
- 273: type …in somata and feet…;
- 293: type …cells at early stages…;
Response
All these changes have been done in the revised version.
Reviewer 2 Report
The work is interesting. The authors studied by immunohistochemistry the retinogenesis of quail
(Coturnix coturnix). The paper is well written and well articulated. I would suggest shortening the discussion and adding the conclusion. Also, the figures are unintuitive on first viewing, and the inscriptions inside are poorly visible. The paper is acceptable overall.
1. Do you think the topic is original or relevant to the field? Does it addresses a specific gap in the field?
In my opinion it is a new and interesting topic, appreciable for originality.
2. Include any additional comments on the tables and figures.
Figures are numerous but not very intuitive especially cannot read the interior lettering
Author Response
The work is interesting. The authors studied by immunohistochemistry the retinogenesis of quail
(Coturnix coturnix). The paper is well written and well articulated. I would suggest shortening the discussion and adding the conclusion . Also, the figures are unintuitive on first viewing, and the inscriptions inside are poorly visible. The paper is acceptable overall.
- Do you think the topic is original or relevant to the field? Does it addresses a specific gap in the field?
In my opinion it is a new and interesting topic, appreciable for originality.
- Include any additional comments on the tables and figures.
Figures are numerous but not very intuitive especially cannot read the interior lettering.
Response
We want to thank reviewer 2 for the advice. Figures have been improved and a general conclusion has been added to the revised version of the paper.